# Delayed detection of leprosy cases: A systematic review of healthcare-related factors

Yudhy Dharmawan[1,2]*, Ahmad Fuady[1,3,4], Ida J. Korfage[1], Jan Hendrik Richardus[1]

1 Department of Public Health, Erasmus MC, University Medical Center Rotterdam, Rotterdam, The Netherlands, 2 Faculty of Public Health, Universitas Diponegoro, Semarang, Indonesia, 3 Department of Community Medicine, Faculty of Medicine, Universitas Indonesia, Jakarta, Indonesia, 4 Primary Health Care Research and Innovation Center, Indonesian Medical Education and Research Institute, Faculty of Medicine Universitas Indonesia, Jakarta, Indonesia

* y.dharmawan@erasmusmc.nl, yudhydharmawan@lecturer.undip.ac.id

**Data Availability Statement:** All relevant data are within the manuscript and its Supporting Information files.

**Funding:** This work was done as part of a PhD scholarship in Health Sciences at Erasmus MC,

## Abstract

### Background

In new leprosy cases, grade 2 disability (G2D) is still a public health burden worldwide. It is often associated with the delayed leprosy diagnoses that healthcare systems should play a crucial role in preventing. The aim of this systematic review was to identify healthcare factors related to delays in case detection in leprosy.

### Methods

PRISMA (Preferred Reporting Items for Systematic Reviews and Meta-Analysis) was used as a guideline in this research. The study protocol was registered in the PROSPERO (International Prospective Register of Systematic Reviews) with reference code CRD42020189274. Data was collected from five electronic databases: Embase, Medline All Ovid, Web of Science, Cochrane CENTRAL, and the WHO Global Health Library.

### Results

After applying the selection criteria for original empirical studies, and after removing duplicates, we included 20 papers from 4313 records. They had been conducted in ten countries and published between January 1, 2000, and January 31, 2021. We identified three categories of healthcare factors related to delayed case. 1) Structural factors, such as i) financial and logistic issues, and geographical circumstances (which we classified as barriers); ii) Health service organization and management including the level of decentralization (classified as facilitators). 2) Health service factors, such as problems or shortages involving referral centers, healthcare personnel, and case-detection methods. 3) Intermediate factors, such as misdiagnosis, higher numbers of consultations before diagnosis, and inappropriate healthcare services visited by people with leprosy.

University Medical Center Rotterdam, that was generously provided by Universitas Diponegoro in Indonesia to YD. The funder of this scholarship played no role in the study design, data collection, data analysis, decision to publish, or preparation of the manuscript.

**Competing interests:** The authors have declared that no competing interests exist.

## Conclusions

Delays in leprosy case detection are due mainly to misdiagnosis. It is crucial to improve the training and capacity of healthcare staff. To avoid misdiagnosis and reduce detection delays, national leprosy control programs should ensure the sustainability of leprosy control within integrated health services.

### Author summary

New leprosy patients diagnosed with visible physical deformities represent a significant disease burden that also poses an important public health challenge. The physical deformities often result from long delays in case detection. Greater insight into the healthcare factors that contribute to such delays will support the development of effective prevention programs. We therefore reviewed all studies on the healthcare factors related to case-detection delay that had been published between January 1, 2000, and January 31, 2021. Twenty studies were included in our analysis. We found that misdiagnosis was a core healthcare factor related to delays. Other common factors included inappropriate health services, a high number of consultations before diagnosis; and a lack of referral centers, healthcare personnel, and case-detection methods. Detection delay was further influenced by geographical circumstances, financial and logistic issues, and health-service organization and management including the level of the decentralization of healthcare programs. Because delays in leprosy case detection are due mainly to misdiagnosis, we recommend to improve the training and capacity of healthcare staff. To avoid misdiagnosis and reduce detection delays, national leprosy control programs should ensure the sustainability of leprosy control within integrated health services.

## Introduction

Over recent decades, leprosy-control programs have substantially reduced the number of new leprosy cases worldwide [1]. The process started in the 1980s, when the introduction of multi-drug therapy (MDT) was combined with nationwide elimination programs that involved health education, case-detection campaigns, and improvements in the quality of leprosy-related services [1,2]. Unfortunately, however, leprosy remains a public-health problem, mainly because the nerve damage it causes often leads to irreversible secondary impairments of the hands, feet, and eyes [3]. Visible impairments and disabilities such as claw hand, foot drop, and lagophthalmos are classified by the World Health Organization (WHO) as leprosy grade 2 disability (G2D) [4,5].

With 10,816 new cases diagnosed in 2019, G2D leprosy now represents a considerable disease burden [6]. It is often associated with delays in diagnosis and appropriate treatment [5,7,8]: the percentage of new G2D cases correlates clearly with delays in case detection–and the longer the delay, the higher the percentage with G2D [9]. The WHO therefore views the rate of G2D cases as an indicator of the quality of the leprosy health services: if the G2D rate is high, it is a clear indication that these services need to be improved, especially with regard to detection [8,10,11]. Relative to the 2020 baseline rate of 1.30 per million population with G2D, the WHO's Leprosy Global Strategy aims for a 90% reduction of G2D in new leprosy cases by 2030 [6].

Detection delays may have many causes. In our recent systematic review identifying the individual and community factors determinants of these delays [9], we concluded that interventions should focus on health-service-seeking behavior and should also consider relevant individual, socioeconomic, and community factors, including stigmatization. However, healthcare-related factors are also important, such as an inability to diagnose leprosy, and health staff's inability to recognize its early signs and symptoms [6,12]. National healthcare systems, including referral centers and primary care services, have a crucial role in preventing the resulting delays [13]. Improved quality of care and interventions to reduce detection delays depend on the systematic identification of healthcare-related factors to decrease leprosy disability in high and low endemic countries, which is what this systematic literature review aimed to achieve.

## Methods

The protocol for this review was based on the Preferred Reporting Items for Systematic Reviews and Meta-Analyses (PRISMA) guidelines for systematic reviews and meta-analyses [14]. The study protocol was registered in the International Prospective Register of Systematic Reviews (PROSPERO) under reference code CRD42020189274.

### Selection criteria and search

Our search for delayed leprosy case detection was based on (a) the period of delay calculated from the beginning of signs or symptoms to diagnosis, either in numerical or categorical values; and (b) the occurrence of G2D. To identify factors determining delayed detection, we used PICO (Population, Intervention/Exposure, Comparison and Outcome) to make structured inclusion and exclusion criteria. Population: leprosy patients in leprosy high and low endemic countries were included. Exposure: Health care services factors. Comparison: we included all studies which presented case detection delay as expressed in time periods of months and as cases categorized as G2D without comparing the health care services factors that contributed to them. Outcome: the disability in leprosy. We systematically searched five electronic databases: Embase, Medline All Ovid, Web of Science, Cochrane CENTRAL, and the WHO Global Health Library (for details of the search strategy, see S1 Text). We included original empirical leprosy-related studies published in English between January 1, 2000, and January 31, 2021. We excluded case reports, articles without full text (abstract only), and articles that made no reference either to delayed case detection or to factors associated with it.

### Data extraction

To select articles for full-text screening, two reviewers (YD and AF) independently screened article titles and abstracts. Data from articles were extracted and double-entered into Microsoft Excel. Disagreements were settled by a third reviewer (IK or JHR). The extracted data included author(s), year of publication, article title, journal title, study design, study setting, number of study participants, and summaries of the relationship between healthcare factors and delays in leprosy case detection.

### Data analysis

After reviewing all the possible healthcare factors, we listed them as variables associated with delays in leprosy case detection. These variables were then categorized into three main groups: 1) Structural factors; 2) Health service factors; and 3) Intermediate factors by considering causal pathways written in each study reviewed. In the quantitative studies, the strength of the

association between factors was expressed as Odds Ratios (ORs), adjusted Odds Ratios (aORs), Hazard Ratios (HRs), and/or significance p-values. In the qualitative studies, we summarized the content of the studies that stated a relationship between healthcare factors and delayed leprosy case-detection or G2D. We conducted a narrative analysis. A meta-analysis was not feasible due to the variation in the methods as applied in the included studies. The methods and results have been reported following the PRISMA guidelines (see S1 Table).

### Evaluation of the quality of studies

The quality of articles was assessed using a risk-of-bias instrument for potential study-design biases. For quantitative studies, a scoring checklist was used to assess the quality of the research hypothesis, the study population, selection bias, exposure, outcome, and confounding; and to formulate an overall opinion of the study's validity and applicability through the accumulation score of the assessment of the method, sampling, data, result, ethics, implications, and usefulness of the study. This assessment was used for all papers that have been reviewed, which all have observational study design [15]. For qualitative studies, a COREQ checklist was used to evaluate the research team and reflexivity; the study design; and our analysis and findings [16]. For mixed-method studies, both methods were combined. Quality was evaluated by two reviewers (YD and AF). In cases of disagreement, a third reviewer (IK or JHR) was invited to resolve the issue.

### Results

In the five databases we identified 7048 articles published between January 2000 and January 2021. After removing duplicates, 4313 articles remained. After the titles and abstracts had been screened, 65 full papers were assessed for eligibility. In the final stage, after the full-text screening, 20 studies were included for analysis. Fig 1 shows the flowchart of article selection according to the PRISMA guidelines.

Twelve articles were observational studies with quantitative analysis [17–28]: nine cross-sectional studies [17,18,21–27], one pre–post study [20], one longitudinal cohort [19], and one retrospective study [28]. Two articles used qualitative analysis [29,30], and six used mixed-methods analysis [31–36]. Fourteen studies (70%) collected data through interviews [17,18,21,23,24,26,28–32,34–36]; while five studies assessed delayed case-detection by reviewing medical records [19,22,25,27,33]; and one assessed delayed case detection through registered data [20]. Most studies were conducted in Asia (five in China, three in India, two in Nepal, one in India and Bangladesh, and one in Myanmar), followed by seven studies in South America (four in Brazil and one each in Colombia, Peru, and Paraguay), and one study in the United Kingdom. Studies were done in various settings: community (n = 7) [17,18,23,27,31,34]; hospital (n = 5) [19,25,29,32,33]; clinic (n = 5) [20–22,30,35]; *hospital* and clinic combined (n = 1) [26]; and clinic and regions/states combined (n = 1) [36]. One was a nation-wide data assessment (n = 1) [24]. As well as assessing the experiences of leprosy patients, three studies also involved healthcare professionals [29,35,36]; one study involved pastors [29]; and one study involved the parents of leprosy patients [31]. Detailed information on the selected studies is given in Table 1.

Fig 2 provides a framework for the factors showing their interrelationships and their contribution pathways to delayed case detection. Most factors and their interactions were based on the literature-review findings, supplemented with the interaction between some factors that are inferred to be based on plausibility. This interaction is indicated by the color code in the figure.

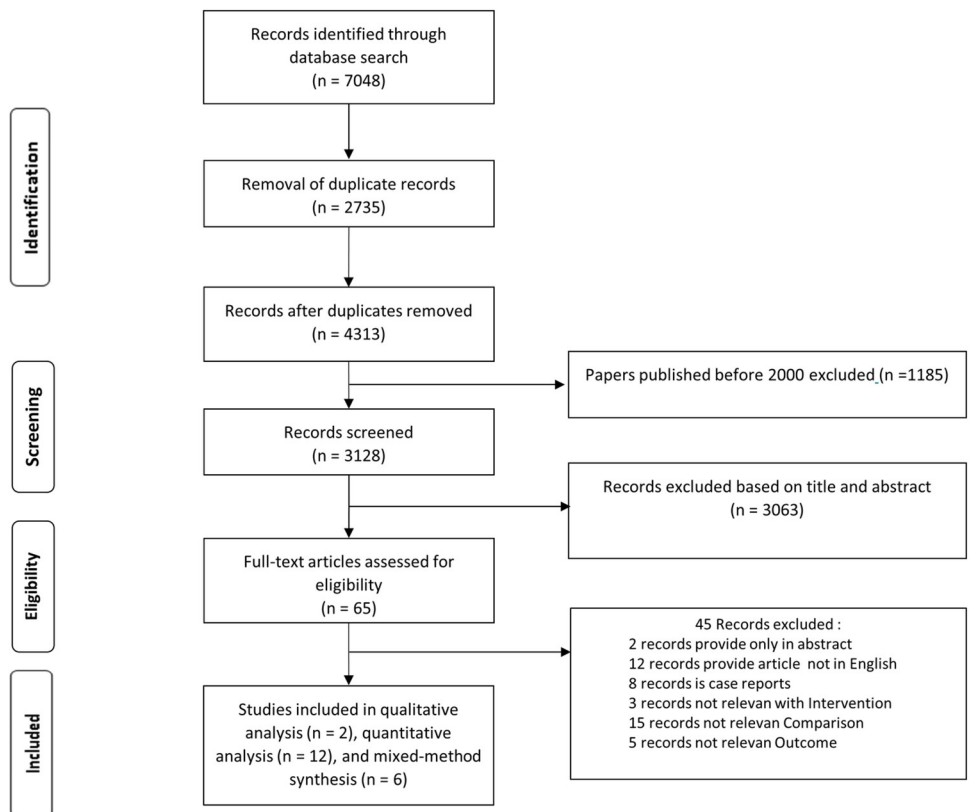

**Fig 1. Flow diagram of the selection process.**

## 1. Structural factors

Structural factors are basic health-service-related factors that contribute to delays in case detection. They include i) financial and logistic issues, and geographical circumstance (which we classified as barriers); ii) health service organization and management including of the level of decentralization of healthcare services (classified as facilitators).

Two studies in Peru and Brazil [30,35] both published in 2020 reported delays related to poor geographical access to healthcare services. Another barrier described by one study post-elimination in Myanmar (2020) was related to lack of financial and logistic support, which limited community-based leprosy field activities [36]; health staff faced high transportation costs for supervising the leprosy control program, and relied on other health programs for field activities [36]. One study in Peru (2010) that evaluated pre and post the leprosy control program decentralization reported that the decentralization of leprosy control and its integration into primary healthcare was a facilitator that led to significantly earlier case detection (p = 0.012) [20]. Decentralization led people with leprosy to visit health services closer to their home with percentages increasing from 23·5% to 46·6% (p<0.001), which led in turn to an increase in self-reporting from 38·7% to 51·1% (p = 0.010) and to a reduction of G2D leprosy at diagnosis from 16.2% to 7.6% (p = 0.012). In contrast, disruptions of decentralized leprosy care, such as a high turnover of health staff and low commitment, contributed to longer case-detection intervals [35], as reported in one study in Brasil (2020) that aimed to analyze assistance provided to people affected leprosy by healthcare management. One paper reported [32] that, in Nepal, the implementation of fully integrated leprosy services contributed to earlier

**Table 1. Study characteristics of the included papers.**

| First author, year of publication | Study design | Country | Setting | Sample size |
|---|---|---|---|---|
| **Quantitative study design** | | | | |
| **Libardo Gomez, 2018 [17]** | Cross-sectional study based on interviews | Colombia | Community | 249 |
| **Furen Zhang, 2009 [23]** | Cross-sectional study based on interviews | China | Community | 88 |
| **Peter Nicholls, 2005 [18]** | Cross-sectional study based on interviews | India | Community | 356 |
| **Cacilda Da Silva Souza, 2003 [26]** | Cross-sectional study based on interviews | Brazil | Hospital, Clinic | 40 |
| **Mary Henry, 2016 [21]** | Cross-sectional study based on quantitative questionnaires | Brazil | Clinic | 122 |
| **XS Chen, 2000 [24]** | Cross-sectional study based on patient records | China | National | 27,928 |
| **Jin Lan Li, 2016 [27]** | Cross-sectional study based on patient records | China | Community | 1274 |
| **Liu Jian, 2017 [22]** | Cross-sectional study based on patient records | China | Clinic | 65 |
| **Diana Lockwood, 2001 [25]** | Cross-sectional study based on case-note review | UK | Hospital | 28 |
| **Tongsheng Chu, 2020 [28]** | Observational; retrospective study | China | Community | 232 |
| **Peter Nicholls, 2003 [19]** | Observational; by patient cohort | Bangladesh, India | Hospital | 2664 |
| **Priscila Fuzikawa, 2010 [20]** | Pre- and post-analysis of the decentralization of leprosy-control activities. | Brazil | Clinic | 435 |
| **Qualitative study design** | | | | |
| **Peter Nicholls, 2003 [29]** | A participatory method based on semi-structured interviews, focus groups, observation, and free listing | Paraguay | Hospital | 36 |
| **Carmen Osorio-Mejia, 2020 [30]** | A qualitative method based on semi-structured interviews | Peru | Clinic | 30 |
| **Mixed-method design** | | | | |
| **Thirumugam Muthuvel, 2017 [31]** | A quantitative component based on a matched case-control design with interviews, followed by a descriptive qualitative component | India | Community | 210 |
| **Sonia Raffe, 2013 [32]** | Quantitative component based on a cross-sectional approach. Qualitative data were collected from semi-structured interviews with patients, case-notes review, and brief clinical examinations | Nepal | Hospital | 78 |
| **Sachin Ramchandra Atre, 2011 [34]** | A quantitative component based on a cross-sectional descriptive and qualitative design with semi-structured interviews | India | Community | 58 |
| **Ulla Britt Engelbrektsson, 2019 [33]** | A quantitative and qualitative method based on interviews and review of patients' documents | Nepal | Hospital | 81 |
| **Cavalcante, 2020 [35]** | The program's municipal coordinator provided the quantitative data on the notified cases, and the qualitative data were obtained by semi-structured script | Brazil | Clinic | 19 |
| **Myo Ko Ko Zaw, 2020 [36]** | The quantitative analysis used an ecological study design for regional data aggregation, and the qualitative data were collected by interview | Myanmar | Regions/states, Clinic. | 42 |

case detection than the earlier non-integrated services, when people visited specialist services as only option for care, often leading to long delays. The low endemicity of leprosy can also contribute to delays in detection: one qualitative study with mix method from India that was done in three districts in one state [31] reported increased detection delays and a lack of involvement by health staff in the National Leprosy Elimination Program (NLEP) after leprosy was declared to have been "eliminated" or "controlled" by the central government.

Fig 2 includes a further structural factor: health-service organization and management. Although this umbrella term was not mentioned explicitly in the articles reviewed, there are many allusions to it.

## 2. Health services factors

Our review found many Health service factors related to healthcare and human resources that contributed to delayed case detection (Fig 2). One study reported that the limited number of

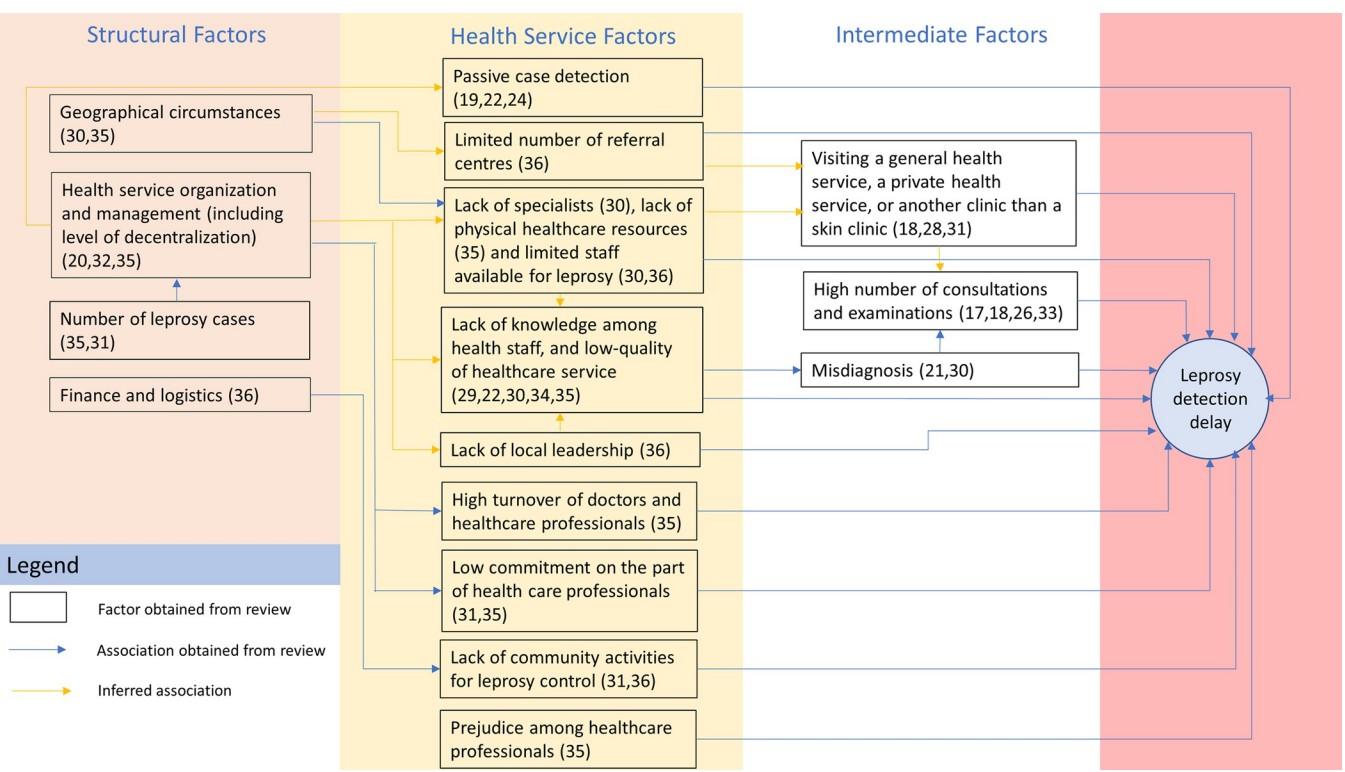

**Fig 2. The framework of healthcare factors related to leprosy case-detection delays.**

referral centers for providing appropriate leprosy care [36] was related to such delays. Regarding personnel resources, three studies [30,35,36] reported that the limited availability of healthcare professionals contributed to delays. One study [30] reported that a lack of specialists caused by geographical barriers was associated with delays, as a result of which people with leprosy perceived health services for leprosy to be limited and geographical access to them difficult. One paper [36] stated that, after the completion of leprosy-elimination programs, many members of health staff mentioned that delays had been related to a lack not only to of local leadership, but also of interest in and commitment to leprosy. Another qualitative study [35] mentioned that high personnel turnover, lack of physical resources, and healthcare professionals' low commitment could contribute to detection delays. The same study also reported that prejudice about leprosy among basic healthcare professionals often led to unnecessary referrals and thereby delays [35].

Regarding community perceptions and activities in the leprosy-care services, several studies reported leprosy patients' perceptions of health workers' knowledge and skills and the quality of health services [22,29–31,35]. One qualitative study [31] reported on the important role played by community volunteers in the public health sector, whose involvement helped reduce delays through the early identification and referral of patients to a health facility.

Two studies reported that lack of knowledge regarding leprosy reactions (a complication of leprosy) were an indication of health staff's unfamiliarity with leprosy [22,34]; as staff had difficulty diagnosing leprosy, they contributed to detection delays [22]. Two studies investigating health staff's diagnostic capabilities revealed patients' perceptions that health staff did not know what leprosy was [29], and described health professionals at different levels of the healthcare system as *"remarkably unprepared to diagnose leprosy"*[35]. Two qualitative studies

reported patients' feelings about the cost of health services, their fear of the diagnosis, the uncertainty of treatment [29], and the mistrust they encountered at the public health facilities [31]. Speaking about the health services, patients referred to a *"lack of compassion among doctors"* and *"long waiting [times]"*; and said *"They don't treat us good in the government hospital"*, *"I don't like to go to Public Health Centers"*, and *"doctors should be kind to patients"* [31].

**Case-detection method.** Three of the four studies [19,22–24] investigating the role of case-finding methods reported a statistically significant association of these methods with delayed case detection. One quantitative study showed that the shortest detection delay was produced by a contact survey in which each household member of every leprosy patient detected had a physical examination (p<0.001) [19]. One cross-sectional study based on patient records found that household contact survey yielded fewer new patients with a disability than all of the following: a suspect survey (medical examination of people who are suspected of having leprosy); self-reporting (by patients), attending a general skin clinic, and spot surveys (surveys for suspected spots on the skin as a clue to leprosy) (p<0.0001) [22]. One cross-sectional study based on patient records found that detection delays with contact survey (household examination) was shorter by 18 months and notification by skin clinics shorter by 18 months than self-reporting (23 months), screening (25 months), and clue surveys, also known as spot surveys (30 months) (p<0.01) [24]. Although another cross-sectional study based on interviews [23] found no association between mode of leprosy detection and total delay in diagnosis, it provided no p-values.

## 3. Intermediate factors

We identified three intermediate factors that contributed to delayed case detection (Fig 2). These factors occur in a causal pathway from health service factors to case detection delay. The first was misdiagnosis leading to mistreatment. Misdiagnosis was often cited as a factor in detection delays [17,21,22,26,29,30,32,36]. The misdiagnosis that was reported most was arthritis, photosensitivity, nerve disease or other skin diseases [32]. One study [26] reported that for periods lasting between three months and 10 years before being diagnosed with leprosy, 32.5% of respondents had been living with a diagnosis other than leprosy. Two studies reported that detection had been delayed in 70% [17] and 65% [32] of their patients.

One study [19] reported that patients needing treatment for leprosy reactions (a severe complication of leprosy that often leads to nerve damage) had had shorter delays at the first intake than those without leprosy reactions (p<0.01). This was because leprosy reactions often cause severe acute symptoms such as pain, swelling of patches, and fever, which drove patients to seek medical attention quickly.

Misdiagnosis can occur because of non-specific symptoms of leprosy, or even of being asymptomatic in the initial stage of disease, and because of healthcare staff's low awareness of the initial appearance of leprosy. A study from the UK [25] reported that 6 out of 12 leprosy cases came with unusual presentations, such as pure neuritic leprosy, trigeminal nerve involvement, and anesthetic patches on the thigh, and that these led to misdiagnosis. Another study from Brazil [17] reported that 4.8% of leprosy cases were asymptomatic and diagnosed accidentally.

The second intermediate factor in detection delays was the number of consultations and examinations–physical and otherwise–that were needed before leprosy was diagnosed. Six studies investigated the relationship between delayed case detection and the number of examinations and/or consultations in healthcare services [17,18,21,23,26,33]. On average, the detection delay for patients who needed five or more consultations was 24.4 months longer than for those diagnosed during the initial consultation (p = 0.009) [17]. Repeated contact with health-

service practitioners was associated with delayed detection (OR: 2.05, 95% CI: 1.42–2.97, p<0.001) [18]. For every additional examination, the odds of a patient that would receive a quicker diagnosis of leprosy were 46% (OR: 0.539, 95% CI: 0.393–0,739, p<0.001) [21]. One study reported that many patients (45.5%) had been diagnosed at their second to fifth visit [23]. Another study reported that 30% of patients in its sample of cases had needed more than five visits to medical services before the diagnosis was suspected [26]. The delay for those with 1–3 visits was less than it was for those with four or more visits (p = 0.051) [33]. In contrast, two studies [17,21] reported that examinations confirming the diagnosis were not associated with detection delay.

The third intermediate factor in detection delays related to the type of healthcare service. Lack of referral services or the existence of integrated leprosy services could lead patients to visit nearby health services that had only limited experience of diagnosing leprosy. Three studies [18,28,31] reported statistically significant relationships between delayed case detection and the type of healthcare service patients had visited. Delayed case detection was associated with a first consultation at a general health service (aOR:2.10, 95% CI: 1.46–3.00, p<0.001) [18]; at another service than a skin clinic (OR: 3.21, 95% CI: 1.68–6.14, p< 0.001) [28]; and at a private healthcare provider (aOR:2.6, 95% CI: 1.4–5.2, p<0.05) [31]. One study [21] reported that health system delays were not significantly associated with referral to another doctor (OR: 1.667, 95% CI: 0.8–3.4, p = 0.176).

## Discussion

This systematic review distinguished three categories of healthcare factors related to delayed case detection of leprosy: 1) Structural factors, which included geographical circumstance, financial and logistic issues, and health-service organization and management including the level of decentralization; 2) health service factors, which included healthcare personnel, case-detection methods, and lack of referral centers or issues with them; and 3) intermediate factors such as misdiagnosis, a high number of consultations, and inappropriate healthcare services visited by people with leprosy.

Misdiagnosis is a key proximal factor related to delays in case detection. Due to the wide variation of signs and symptoms, leprosy is often not easily recognized. While hypopigmented or reddish skin lesions with central hypoesthesia are common presenting skin symptoms, many patients present primarily with signs of peripheral neuropathy [37]. As leprosy is diagnosed on the basis of clinical, microbiological, and histopathological features, diagnosis is difficult for general healthcare professionals [38]. If it is not diagnosed immediately, people with leprosy will need more visits to healthcare providers. Longer detection delays can also be caused by wrong diagnoses [39]. Health staff's ability to diagnose leprosy plays a vital role in detecting leprosy at an early stage [34,40]. Bad experiences with diagnosis and treatment by healthcare professionals will lead to a mistrust of healthcare services, with patients thinking that the leprosy healthcare services are a waste of time and money [31].

While training and capacity building has been shown to increase health workers' capabilities with regard to the early diagnosis and control of leprosy [31,41], any capacity for leprosy detection can be disrupted by high levels of staff mobility and turnover [42]. Maintaining skill levels in leprosy-control programs is also challenging, and there is still a risk that healthcare staff will lose their awareness of the need for early case detection [42]. As periodic efforts should be made to engage and capacitate all healthcare staff [8], training content should be tailored to their needs and skills. This content should also be integrated into health programs, and include aspects of leadership and management [43,44]. To maintain a high awareness of leprosy, improve diagnostic skills, and avoid professionals' prejudice with regard to diagnosing

leprosy, training must be implemented at different levels, including that of specialists such as dermatologists and neurologists [23].

Accessible leprosy health services and referral hospitals are essential to reducing delays in detection [45]. Lack of dedicated healthcare services and laboratory facilities will lead to misdiagnoses and further delays. National leprosy control programs must ensure the availability of healthcare for leprosy patients, with competent staff and minimal staff turnover [46]. National leprosy control programs should also provide sufficient regulations and incentives for maintaining the readiness of health staff to provide proper diagnosis and treatment [28,47]. As case detection is sometimes delayed by visits to private clinics and by a lack of community activities such as active leprosy case-finding surveys, leprosy control programs could adopt approaches developed by tuberculosis programs to find the cases as early as possible, such as a public-private mixed (PPM) approach and the involvement of community health workers in the national leprosy control program. The leprosy control program can create health care network services in one district/city involving all public and private health facilities of leprosy coordinated by the District/City Health Office to increase the leprosy early case detection [48,49].

With regard to national programs, it should be understood that a key component–which also helps reduce detection delays–is sustainability [50]. To achieve the sustainability brought by maintaining these programs and ensuring their financing over time, many leprosy services have now been integrated into essential health and primary care [40]. Early detection would be aided by a strategy of decentralizing and integrating leprosy detection and treatment into general health centers and community programs [26,33,51]. To overcome any lack of specialists, health staff, and referral hospitals, especially in areas with geographic barriers, national leprosy control programs should support their own decentralization and the integration of leprosy control programs into primary healthcare [30,35]. These programs should also take care not to declare prematurely that leprosy has been eliminated, as this can reduce healthcare professionals' awareness of and commitment to the disease, thereby ending its early detection [31,36].

There are various methods for detecting leprosy in the community: household contact surveys, suspect surveys, self-reporting, general skin-clinic examinations, spot surveys, screening, rapid village surveys, and clue surveys [24]. National leprosy control programs should encourage active case finding–especially through household contact surveys–as an effective strategy mainly for targeting high endemic areas (hot spots) in individual districts [8,31]. An effective and feasible intervention recommended by the WHO for reducing leprosy cases is to follow a contact survey by providing post-exposure prophylaxis to eligible contacts with single-dose rifampicin (SDR). This intervention can prevent delay effectively due to the leprosy control program must contact the community directly to increase the awareness of the community through health promotion efforts and do active case finding [52–54].

This article is the first systematic review on healthcare factors related to delays in leprosy case-detection. While it summarizes the factors determining delayed leprosy case detection on the health-system supply side, it also has three main limitations. First, because methods and research settings differed between studies, it is not easy to generalize findings. Second, as we included only studies in English, we could not capture other-language publications that may have originated in countries with a high leprosy burden, such as Indonesia and Brazil [12]. Third, the various definitions used in the different studies made it difficult to achieve standardization.

## Conclusion

Misdiagnosis is the main factor in healthcare-related delays in leprosy case-detection. Interventions should focus on avoiding misdiagnosis of leprosy. National leprosy control programs

must prioritize the training and capacity building of their healthcare staff. To avoid misdiagnosis and reduce detection delays, the sustainability of leprosy control within integrated health services should be ensured.

## Supporting information

**S1 Table. PRISMA Checklist.**
(DOC)

**S1 Text. Search strategy for each database.**
(DOCX)

## Acknowledgments

The authors thank Wichor Bramer, Sabrina Meertens-Gunput, Elise Krabbendam, Maarten Engel, and Christa Niehot at the Erasmus MC Medical Library for developing and updating our search strategies.

## Author Contributions

**Conceptualization:** Yudhy Dharmawan, Ida J. Korfage, Jan Hendrik Richardus.

**Data curation:** Yudhy Dharmawan.

**Formal analysis:** Yudhy Dharmawan, Ahmad Fuady.

**Investigation:** Yudhy Dharmawan.

**Methodology:** Yudhy Dharmawan, Ida J. Korfage, Jan Hendrik Richardus.

**Supervision:** Ahmad Fuady, Ida J. Korfage, Jan Hendrik Richardus.

**Validation:** Yudhy Dharmawan, Ahmad Fuady, Ida J. Korfage, Jan Hendrik Richardus.

**Visualization:** Yudhy Dharmawan.

**Writing – original draft:** Yudhy Dharmawan.

**Writing – review & editing:** Yudhy Dharmawan, Ahmad Fuady, Ida J. Korfage, Jan Hendrik Richardus.

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
