## [Decision Letter · Decision Letter 0]

14 Jun 2022

Dear Mr Dharmawan,

Thank you very much for submitting your manuscript "Delayed detection of leprosy cases: A systematic review of healthcare-related factors" for consideration at PLOS Neglected Tropical Diseases. As with all papers reviewed by the journal, your manuscript was reviewed by members of the editorial board and by several independent reviewers. In light of the reviews (below this email), we would like to invite the resubmission of a significantly-revised version that takes into account the reviewers' comments. 

We cannot make any decision about publication until we have seen the revised manuscript and your response to the reviewers' comments. Your revised manuscript is also likely to be sent to reviewers for further evaluation.

Sincerely,

Alberto Novaes Ramos Jr

Associate Editor

Godfred Menezes

Deputy Editor

Reviewer's Responses to Questions

**Key Review Criteria Required for Acceptance?**

**Methods**

-Are the objectives of the study clearly articulated with a clear testable hypothesis stated?

-Is the study design appropriate to address the stated objectives?

-Is the population clearly described and appropriate for the hypothesis being tested?

-Is the sample size sufficient to ensure adequate power to address the hypothesis being tested?

-Were correct statistical analysis used to support conclusions?

-Are there concerns about ethical or regulatory requirements being met?

Reviewer #1: A well written article

Reviewer #2: -Are the objectives of the study clearly articulated with a clear testable hypothesis stated?

Yes. The aim of this systematic review was to identify healthcare factors related to delays in case detection in leprosy.

-Is the study design appropriate to address the stated objectives?

No. The framework of healthcare factors related to case detection delay(Figure 2.) is confusing. The authors classified healthcare factors related to case detection delay as macrolevel factors, microlevel factors and intermediate factors. It is innovative, however, it is difficult to classify them. For example, in macrolevel factors: iii) health service organization and management (classified as factors underlying the microlevel factors). It is contradictory. The authors believed that poor geographical access is macrolevel factor, but lack of specialists caused by geographical barriers is microlevel factor. The authors believed that decentralization and integration is macrolevel factor but the limited number of referral centers is microlevel factor. What is intermediate factor? it is also confusing. The problem analysis tree is not clear. In my opinion, the quality of healthcare is ususal evaluated by availability and accessibility, and then find their related factors. 

-Is the population clearly described and appropriate for the hypothesis being tested?

Yes.

-Is the sample size sufficient to ensure adequate power to address the hypothesis being tested?

Yes.

-Were correct statistical analysis used to support conclusions?

Yes.

-Are there concerns about ethical or regulatory requirements being met?

Yes.

Reviewer #3: In row 98 – did the authors searched the grey literature? If not, please provide the reason why.

In rows 98-106: It is important to refer with details to the PICO\\PECO criteria (population, exposure, comparison group and outcome) when describing the selection criteria in the text. For example, if delays are measured in days, the authors used any cut-off point to define delay or just used delay as a continuous variables? Would both types be included in the SR? In addition, there if no information on the comparison group or the exposure.

Rows 105-106- Other narrative or systematic reviews were included in the SR? 

In rows 116-124– The authors do not report metanalysis. This was consider in the study and it was not possible or it was not considered in the first place? In addition, please state that a narrative review of the literature was performed.

In rows 128-130 – The sentence is not clear. Also, please provide more details on how the tool to evaluate quantitative studies formultae an opinion of the study validity and applicability. Also, it is a similar tool to all types of quantitative studies (randomized trials and observational studies)?

**Results**

-Does the analysis presented match the analysis plan?

-Are the results clearly and completely presented?

-Are the figures (Tables, Images) of sufficient quality for clarity?

Reviewer #1: Yes, analysis is done as per plan

Reviewer #2: -Does the analysis presented match the analysis plan?

Yes. 

-Are the results clearly and completely presented?

No. The analysis plan is illogical.

-Are the figures (Tables, Images) of sufficient quality for clarity?

No. Figures 2. is confusing.

Reviewer #3: Figure 1 (flowchart) must provide the reasons for excludion and the number of articles excluded by each reason.

In table 1, in the last column that says Sample size (responde rate), I suggest removing response rate as that means something else. If the authors do not know how many individuals were targeted and how many actually were interviewed, qit is impossible to know the responde rate.

Figure 2 is very good but it is unclear, for example, how number of leprosry cases could lead to descentralization. Also, it is unclear what prejudice among healthcare professionals mean – its prejudice of healthcare professionals towards patients?

Row 162-163 – information repeated from methods section

Rows 171-176 – it is confusing the definition of macrolevel factors that seem to act as sometimes barriers and sometimes facilitators. I think reframing the names so to go all to barriers or all to facilitators will clarify the interpretation of the text and figure 2. 

Rows 178-193 – this a nice description of studies but lacks more data on the studies (e.g., country, when it was published and in which context) to improve interpretability of findings. For instance, when the authors say that “the low endemicity of leprosy can also contribute rto delays…”, this was done in the entire country of India or in a certain area? Which type of study was that?

In row 195-197 – The sentence “Fig 2 includes a further macrolevel factor: health-service organization and management. Although this umbrella term was not mentioned explicitly in the articles reviewed, there are many allusions to it.” Seems lost in the text. Maybe the authors could explain in the methods that the process of aggregating studies in macrolevels when that was not present was a subjective decision made by the authors. Also, what was performed if the authors classification into levels (macro, micro and intermediate levels) was discordant with the original study?

In rows 233-235 – It is not possible to perform a matanalysis on the role of case-finding methods and with delayed case detection? It seems 3 studies can be combined into a polled analysis.

In rows 250-251 – Similar. It is not possible to perform a matanalysis on misdiagnosis was on leprosy late case detection? In addition, what was the most type of misdiagnosys cited?

In rows 268-270 – What is the causal pathway that the number of examinations

or consultations in healthcare services can delay diagnosys? Maybe a better interpretation is needed. This a signal of bad trained doctors? Lack of laboratories? Or something else? There is any more information associated to the type of clinic (more or less specialized of the first visits, duration of consultations, etc) that could be associated with delays?

**Conclusions**

-Are the conclusions supported by the data presented?

-Are the limitations of analysis clearly described?

-Do the authors discuss how these data can be helpful to advance our understanding of the topic under study?

-Is public health relevance addressed?

Reviewer #1: Conclusions are supported by the data presented

Reviewer #2: -Are the conclusions supported by the data presented?

No. For example, the decentralization and integration of healthcare services (classified as facilitators) has both advantages and disadvantages.

-Are the limitations of analysis clearly described?

Yes.

-Do the authors discuss how these data can be helpful to advance our understanding of the topic under study?

Yes. 

-Is public health relevance addressed?

Yes.

Reviewer #3: In rows 333-334- - the authors cite “leprosy programs could adopt approaches developed by tuberculosis programs, such as a public private mixed (PPM) approach”. Please provide minimum details so people can understand which “approach” is that. Many readers could not be familiar with either leprosy or TB.

Row 352 – It is confusing why the authors are citing this reference in the sentence. The results are similar to study 53?

Rows 355-356 – How WHO post-exposure profilaxy is related to the main objective of your study? Please provide more details on eligibility and how this could contribute to reduce leprosy burden.

**Editorial and Data Presentation Modifications?**

Reviewer #1: Accept

Reviewer #2: Minor Revision.

Reviewer #3: (No Response)

**Summary and General Comments**

Reviewer #1: Overall a good meta analysis article

Reviewer #2: This systematic review is an interesting study addressing on healthcare factors related to delayed detection of leprosy cases. The findings and conclusions are helpful for formulating leprosy control progarams. The method of data analysis need to be clear and logical.

Reviewer #3: I believe this is a very important work, but the methods described suggest that this is more a literature review rather than a systematic review. I suggest to revise PRISMA guideline to improve the methods and results section. The approach to select papers and extract data from them is not very detailed – which type of papers were you looking for in the SR, inclusion and excludion criteria, methods to extract data and approach to summarise data should be all in the methods. As this type of information was not included, readers cannot be sure if all the literature on the topic was included or if led to too much exclusion. In addition, I believe it is a big limitation only including papers in English. Please see detailed comments below:

PLOS authors have the option to publish the peer review history of their article (what does this mean?). If published, this will include your full peer review and any attached files.

Reviewer #1: Yes: SRINIVAS GOVINDARAJULU

Reviewer #2: No

Reviewer #3: Yes: Julia Pescarini
---

## [Decision Letter · Decision Letter 1]

19 Aug 2022

Dear Mr Dharmawan,

We are pleased to inform you that your manuscript 'Delayed detection of leprosy cases: A systematic review of healthcare-related factors' has been provisionally accepted for publication in PLOS Neglected Tropical Diseases.

Best regards,

Alberto Novaes Ramos Jr

Academic Editor

Godfred Menezes

Section Editor

Reviewer's Responses to Questions

**Key Review Criteria Required for Acceptance?**

**Methods**

-Are the objectives of the study clearly articulated with a clear testable hypothesis stated?

-Is the study design appropriate to address the stated objectives?

-Is the population clearly described and appropriate for the hypothesis being tested?

-Is the sample size sufficient to ensure adequate power to address the hypothesis being tested?

-Were correct statistical analysis used to support conclusions?

-Are there concerns about ethical or regulatory requirements being met?

Reviewer #1: ROB tool, what was used? Guidelines were not mentioned or complied with, and the method adopted is not standardized?

As the idea revolves around the healthcare-related factors, A column, mentioning the nature of healthcare facilities in the country-specific, against each report, to get a clear picture is a mandate.

Reviewer #2: yes

Reviewer #3: -Are the objectives of the study clearly articulated with a clear testable hypothesis stated? Yes

-Is the study design appropriate to address the stated objectives? Yes

-Is the population clearly described and appropriate for the hypothesis being tested? Yes

-Is the sample size sufficient to ensure adequate power to address the hypothesis being tested? n/a

-Were correct statistical analysis used to support conclusions? Yes

-Are there concerns about ethical or regulatory requirements being met? n/a

**Results**

-Does the analysis presented match the analysis plan?

-Are the results clearly and completely presented?

-Are the figures (Tables, Images) of sufficient quality for clarity?

Reviewer #1: The framework, which again was deduced is not satisfactory. Looks like Organisational structural flaws and Health services/system flaws, please check; Revise the framework, as the framework is an important finding of your SR. Also, consider omitting the Intermediate factor.

In many places, statistics may be interpreted, with practical relevance.

Reviewer #2: yes

Reviewer #3: -Does the analysis presented match the analysis plan? Yes

-Are the results clearly and completely presented? Yes

-Are the figures (Tables, Images) of sufficient quality for clarity? Yes

**Conclusions**

-Are the conclusions supported by the data presented?

-Are the limitations of analysis clearly described?

-Do the authors discuss how these data can be helpful to advance our understanding of the topic under study?

-Is public health relevance addressed?

Reviewer #1: OK

Reviewer #2: yes

Reviewer #3: (No Response)

**Editorial and Data Presentation Modifications?**

Reviewer #1: (No Response)

Reviewer #2: yes

Reviewer #3: Accept

The paper has substantially improved since the last version. A single comments that are still not clear include:

Page 381-383 of revised manuscript: The following sentence is unclear and does not directly follow the explanation of the sentence just before it: “This intervention can prevent delay effectively due to the leprosy program must make contact with the community directly to increase the awareness of the community through health promotion efforts and do active case finding” . I suggest removing it.

**Summary and General Comments**

Reviewer #1: English grammar could be improved, inn general otherwise OK

Reviewer #2: all my comments were well addressed

Reviewer #3: (No Response)

PLOS authors have the option to publish the peer review history of their article (what does this mean?). If published, this will include your full peer review and any attached files.

Reviewer #1: **Yes: **SRINIVAS G

Reviewer #2: No

Reviewer #3: **Yes: **Julia Pescarini

---

## [Editor Report · Acceptance letter]

31 Aug 2022

Dear Mr Dharmawan,

We are delighted to inform you that your manuscript, "Delayed detection of leprosy cases: A systematic review of healthcare-related factors," has been formally accepted for publication in PLOS Neglected Tropical Diseases.

Best regards,

Shaden Kamhawi

co-Editor-in-Chief

Paul Brindley

co-Editor-in-Chief
